# The Role of Two Heart Biomarkers in IgA Nephropathy

**DOI:** 10.3390/ijms241210336

**Published:** 2023-06-19

**Authors:** Balázs Sági, Tibor Vas, Rita Jakabfi-Csepregi, Zoltán Horváth-Szalai, Tamás Kőszegi, Botond Csiky, Judit Nagy, Tibor József Kovács

**Affiliations:** 12nd. Department of Internal Medicine and Nephrology, Diabetes Center, Clinical Center, Medical School, University of Pécs, 7624 Pécs, Hungary; balazs.sagidr28@gmail.com (B.S.); vas.tibor2@pte.hu (T.V.); botond.csiky@gmail.com (B.C.); judit.nagy@aok.pte.hu (J.N.); 2Fresenius Medical Care Dialysis Center, 7624 Pécs, Hungary; 3Department of Laboratory Medicine, Medical School, University of Pécs, 7624 Pécs, Hungary; ritacsepregi93@gmail.com (R.J.-C.); horvath-szalai.zoltan@pte.hu (Z.H.-S.); koszegi.tamas@pte.hu (T.K.); 4Szentágothai Research Center, University of Pécs, 7624 Pécs, Hungary

**Keywords:** chronic kidney disease, IgA nephropathy, renal function, heart failure, arterial stiffness

## Abstract

Cardiovascular mortality is a leading cause of death in chronic kidney disease (CKD), as is IgA nephropathy (IgAN). The purpose of this study is to find different biomarkers to estimate the outcome of the disease, which is significantly influenced by the changes in vessels (characterized by arterial stiffness) and the heart. In our cross-sectional study, 90 patients with IgAN were examined. The N-terminal prohormone of brain natriuretic peptide (NT-proBNP) was measured as a heart failure biomarker by an automated immonoassay method, while the carboxy-terminal telopeptide of collagen type I (CITP) as a fibrosis marker was determined using ELISA kits. Arterial stiffness was determined by measuring carotid–femoral pulse wave velocity (cfPWV). Renal function and routine echocardiography examinations were performed as well. Based on eGFR, patients were separated into two categories, CKD 1-2 and CKD 3-5. There were significantly higher NT-proBNP (*p* = 0.035), cfPWV (*p* = 0.004), and central aortic systolic pressure (*p* = 0.037), but not CITP, in the CKD 3-5 group. Both biomarker positivities were significantly higher in the CKD 3-5 group (*p* = 0.035) compared to the CKD 1-2 group. The central aortic systolic pressure was significantly higher in the diastolic dysfunction group (*p* = 0.034), while the systolic blood pressure was not. eGFR and hemoglobin levels showed a strong negative correlation, while left ventricular mass index (LVMI), aortic pulse pressure, central aortic systolic pressure, and cfPWV showed a positive correlation with NT-proBNP. cfPWV, aortic pulse pressure, and LVMI showed a strong positive correlation with CITP. Only eGFR was an independent predictor of NT-proBNP by linear regression analysis. NT-proBNP and CITP biomarkers may help to identify IgAN patients at high risk for subclinical heart failure and further atherosclerotic disease.

## 1. Introduction

Chronic kidney disease (CKD) is a global health challenge for health professionals and healthcare providers worldwide [1,2]. IgA nephropathy (IgAN) is the most common primary glomerular disease-, and is the typical CKD for cardiovascular (CV) risk assessment. IgAN is induced by the accumulation of IgA-containing immune complexes in the kidney glomeruli, which initiates a cascade of inflammatory processes that ultimately result in irreversible glomerulosclerosis, which in 25–30% of the patients causes end-stage renal disease (ESRD), a decreased quality of life (QoL), the need for dialysis or transplantation, and risk of early mortality [3].

In recent years, plasma biomarkers, particularly natriuretic peptides, along with imaging studies, e.g., echocardiographic examinations or coronary CT scans, have been suggested to identify patients at high CV risk in CKD. In chronic kidney disease (CKD), arterial stiffness may be a predictor of cardiovascular (CV) mortality and morbidity [4]. In advanced CKD stages, heart failure (HF) is highly prevalent. In CKD patients with HF, the emergence of cardiac fibrosis may be a pathophysiological change [5]. Patients with CKD suffer severe losses in life expectancy and quality of life, which are primarily caused by a high incidence of cardiovascular (CV) events [2]. In cross-sectional studies, natriuretic peptides have been independently linked to current cardiovascular disease (CVD) and potential future CV events [6,7] in prospective CKD cohort studies [8,9,10,11,12,13,14]. The carboxy-terminal telopeptide of collagen type I (CITP) is a marker of myocardial fibrosis (MF) in CKD [15], and serum levels are directly correlated with the amount of collagen type I deposition in the myocardium of hypertensive patients with HF [16]. A rise in serum biomarkers is difficult to assess in the case of kidney disease, as we do not know whether the levels really rise or whether their excretion is only the cause of the difference.

Data in the literature about the association between arterial stiffness, subclinical heart failure (NT-pro-BNP), cardiac fibrosis (CITP), and renal function in CKD are limited. We wanted to determine new biomarkers in connection with the influence of renal function on CV complications in IgAN, such as heart failure and atherosclerotic disease prediction, and detection of any association between NT-proBNP as a marker of heart failure, CITP as a marker of atherosclerosis and myocardial fibrosis, and CKD.

## 2. Results

The study included 90 patients, 50 of whom were male, and who had an average age of 54.9 ± 14.4 years. Patients were divided into two groups based on eGFR (CKD 1-2 (GFR: >60 mL/min) and CKD 3-5 (GFR: <60 mL/min)). There were significant differences between the two groups in terms of age, eGFR, hypertension, diabetes, LVMI, and hemoglobin level in the baseline characteristics (Table 1). There were significantly higher NT-proBNP (*p* = 0.035), cfPWV (*p* = 0.004), and central aortic systolic pressure (*p* = 0.037), but not CITP, in the CKD 3-5 group (Figure 1).

The number of patients with combined biomarker positivity (NT-proBNP and CITP) was significantly higher in the CKD 3-5 group (*p* = 0.035) compared to the CKD 1-2 group (Figure 2). In the case of LVH (n = 42), the number of patients with combined biomarker positivity (NT-proBNP and CITP) was significantly higher (n = 12/42 vs. 3/48; *p* = 0.002) than in the non-hypertrophic group (Figure 3).

When we divided the patients into two groups based on left ventricular diastolic function (no LVDD (n = 51) vs. LVDD (n = 39)), we found that the diastolic dysfunction group had significantly higher central aortic systolic blood pressure (116.8 vs. 128.5; *p* = 0.034), while the peripheral systolic blood pressure, NT-proBNP, and CITP did not differ between the two groups (Figure 4).

NT-proBNP showed a strong negative correlation with eGFR and hemoglobin levels and a positive correlation with the left ventricular mass index (LVMI), aortic pulse pressure, central aortic systolic pressure, and cfPWV. CITP showed a strong positive correlation with cfPWV, aortic pulse pressure, and LVMI by Spearman’s correlation (Table 2).

According to univariate and multivariate regression analysis, the independent predictor factors for NT-proBNP were gender, eGFR, and LVMI. There were no independent confounders in the case of CITP, while all of the exemplified parameters had an impact on the univariate analysis (Table 3 and Table 4). Only eGFR was an independent predictor of NT-proBNP by linear regression analysis.

## 3. Discussion

In the present study, we demonstrated that NT-proBNP and a combination of NT-proBNP and CITP elevation may predict certain cardiac complications in IgAN. NT-proBNP and CITP biomarkers may help to identify IgAN patients at high risk for subclinical heart failure and further atherosclerotic disease. We found an independent association between eGFR and NT-proBNP, but not with CITP, in IgA nephropathy. There was a strong correlation between NT-proBNP, CITP, and aortic PP and PWVcf. In cases of deteriorated renal function, there was significantly higher central aortic systolic pressure.

Our knowledge of the mechanisms underlying the advancement of kidney disease, cardiovascular disease, and all-cause mortality has greatly benefited from measurements of arterial stiffness [17], which was supported by our former examinations as well [18,19].

Ohno et al. found that in CKD patients central blood pressure is a stronger predictor of CV and renal disease outcomes compared with brachial blood pressure, and should be used to guide antihypertensive therapy [20]. Our results support these observations.

In our research, we came to the same conclusion as Uterstellar et al., who identified NT-proBNP as a highly effective indicator of CV prognosis in CKD patients not receiving dialysis [13,14]. They demonstrated that NT-proBNP remains a robust predictor of outcome, notably for eGFR, even after complete adjustment for potential confounders. This observation is significant, as it has previously been claimed [12,21] that the use of this plasma biomarker as an independent CV outcome marker in CKD may be precluded by the substantial association between NT-proBNP and GFR, which is caused by decreased renal clearance of NT-proBNP in advanced CKD. The CARE for HOME study advises laboratory measurement of NT-proBNP against routine echocardiographic investigations because it is less expensive, less challenging, and takes less time than a normal echocardiographic examination [22].

Two earlier Greek studies showed a connection between altered collagen turnover (collagen type I synthesis) and increased aortic stiffness in treated hypertensive patients without left ventricular (LV) hypertrophy [23] as well as a possible link between altered collagen metabolism and peripheral vascular stiffness in chronic HF [24]; however, they did not examine the connection with renal function.

In patients with CKD, particularly those who have HFpEF, Eiros et al. described a link between LVDD and the biomarker combination of high carboxy-terminal propeptide of procollagen type-I (PICP) and low carboxy-terminal telopeptide of collagen type-I to matrix metalloproteinase-1 (CITP:MMP-1) ratio. These results suggest that CKD promotes the development of biomarker-assessed myocardial fibrosis and LVDD in hypertensive patients with HFpEF, and that changes in collagen type I metabolism as measured by the biomarkers PICP and CITP:MMP-1 ratio worsen with the transition to HFpEF in hypertensive patients [15].

We found a positive correlation between CITP and cfPWV, aortic pulse pressure, and LVMI in our cohort, which suggests an association between worsening kidney function and vascular changes, and means that altered collagen turnover (CITP) can be assumed.

Because of the high prevalence of myocardial fibrosis in patients with CKD and the direct correlation that exists between the degree of impairment of kidney function and the severity of myocardial fibrosis [25,26,27], it has been proposed that pro-fibrotic factors or pathways linked to CKD could exist [28].

Additional factors favor NT-proBNP as a plasma biomarker for CV outcome prediction in clinical practice over an echocardiography-based method. First, the cut-off values for NT-proBNP were significantly lower (150 pg/mL) than those found in echocardiographic investigations [29]. Second, little information is currently available about the therapeutic effects of screening echocardiography in people with or without CKD from prospective interventional trials. In contrast, NT-proBNP-directed cardioprotective therapy techniques have been adopted recently as a result of multiple interventional trials that have shown their superiority over clinically advised treatment strategies in both primary and secondary prevention outside the nephrology community [30,31,32,33]. Based on our data, it may be important in IgAN and CKD as well.

Liang et al. proved that systolic dysfunction and LVDD demonstrated mutually augmentative effects on CV mortality, and suggested that cardioprotection for patients with CKD should be prioritized at an early stage along with conventional nephroprotection. Therefore, cardioprotective management should be initiated as early as possible after CKD diagnosis [34].

Uremia causes structural alterations in the heart, including thickening of the intramural arteries, cardio-myocyte hypertrophy, and myocardial fibrosis. In response to the compounding effects of traditional risk factors and CKD-related risk factors, these structural changes together predispose patients to LVDD [35,36,37]. There is solid proof that variations in collagen myocardial metabolism are connected to interstitial fibrosis in CKD, however, vascular remodeling and cardiomyocyte hypertrophy may represent adaptive reactions to pressure and volume stress [38]. Other metabolic changes, including hypovitaminosis D, hyperparathyroidism, and hyperphosphatemia, are particularly important in more advanced stages of CKD and during dialysis [38,39], and might influence heart function only later. The activation of the renin-angiotensin-aldosterone system (RAAS), which may be responsible for cardiac fibrosis and hypertrophy, is another significant component. Angiotensin II and aldosterone can be involved in myocardial cell hypertrophy and fibrosis independent of afterload; however, activation of the intracardiac RAAS appears to be critically engaged in the overload situation observed in dialysis [40].

Unfortunately, we did not examine the effects of RAAS inhibitors in this study. Although we did not examine the role of these factors, these observations may explain the observed differences in the LVDD values of our IgAN patients.

## 4. Materials and Methods

We included 90 renal biopsy-proven IgAN patients in our cross-sectional study, and we analyzed their data retrospectively. The University of Pécs Regional Research Ethics Committee approved the study protocol, and all participants provided written consent to its completion. The inclusion criterion for the study was confirmed IgAN over the age of 18, while exclusion criteria were previous or current immunosuppressive treatment (due to the modifying effects of the biomarker) and severe comorbidities, such as malignancies that required active treatment or acute infection.

At the start of patient enrollment, echocardiography measurements were performed and classic CV risk factors, including hypertension, carbohydrate metabolism disorder, obesity, lipid abnormalities, smoking, and patient medications such as antihypertensive drugs (ACEI/ARB, BB, CCB), and statins were recorded. The obesity inclusion criterion was a BMI over 30 kg/m^2^. The CKD-EPI formula was used to estimate renal function (eGFR, mL/min, 1.73 m^2^). A 24-h blood pressure monitor using Meditech ABPM devices was relied on to determine the patient’s 24 h average systolic and diastolic blood pressure, pulse pressure, and diurnal index. Echocardiography was performed at the start of the study (see below). During these visits, medical events that had occurred since the previous visit were recorded, the patient’s physical status was examined, and detailed laboratory tests were performed. Blood pressure values were determined from the average of three measurements taken after 10 min of rest.

### 4.1. Biomarker Measurement

N-terminal prohormone of brain natriuretic peptide (NT-proBNP) was determined in the accredited Department of Laboratory Medicine (Universiy of Pécs, NAH-9-0008/2021), by a fully automated immonoassay method (Roche^®^ Gmbh, Mannheim, Germany), while carboxy-terminal telopeptide of collagen type I (CITP) was measured using enzyme-linked immunosorbent assay (ELISA) kits (CITP kit by MyBioSource^®^, San Diego, CA, USA) in the Szentágothai Research Center (University of Pécs) following the manufacturer’s protocol.

### 4.2. Arterial Stiffness Measurement

Arterial stiffness was determined by measuring carotid–femoral pulse wave velocity (cfPWV). cfPWV was measured using applanation tonometry (SphygmoCor System; AtCor Medical, Sydney, Australia). Measurements were performed in the morning in the supine position after at least 10 min of rest in a quiet temperature-controlled room. Pulse wave recording was performed consecutively at two superficial artery sites (carotid–femoral segment). The cfPWV was calculated, and central aortic pressure was measured by the device. Augmentation index (Aix) is based on blood pulse-wave reflection, and is an accepted measure of arterial stiffness and risk factor for cardiovascular disease. The Aix is commonly accepted as a measure of the enhancement (augmentation) of central aortic pressure by a reflected pulse wave.

### 4.3. Echocardiographic Measurement

Echocardiography was performed with the Aloka SSD 1400; two operators were involved in the study. Left ventricular mass (LVM) was calculated from 2D images of the left ventricular short-axis muscle area and apical left ventricular length (LVM = (5/6 area × length)). The left ventricular mass index (LVMI) of g/m^2^ was calculated using Devereux’s formula, and the cardiac mass was indicated by lean mass. LVMI was determined based on the Cornell criterion and indexed for height (in meters). The left ventricular ejection fraction (LVEF) was calculated by adding the diastolic and systolic volumes of the left ventricle using the unidirectional Simpson method: EF = (Dvol-Svol)/Dvol × 100. Diastolic function was determined by mitral inflow and pulmonary venous flow based on conventional spectral Doppler measurements. We measured the ratio of the E wave to the A wave (E/A ratio), the isovolumetric relaxation time (IVRT), and the deceleration time of the E wave. LVH was defined as abnormal RWT and/or LVMI.

### 4.4. Statistical Analysis

Statistical analyses were performed using SPSS 21.0 software (SPSS, Inc., Chicago, IL, USA). The Kolmogorov–Smirnov test was used to determine the normality of the data distribution. Non-normally distributed parameters were transformed logarithmically. A comparison of clinical and laboratory parameters was made using Student’s *t*-test and ANOVA as appropriate. The mean SD was used to express data with a normal distribution. Correlations between continuous variables were evaluated with linear regression using the Pearson test, while the Spearman correlation test was used for categorical variables. Values of *p* > 0.05 were considered statistically significant.

## 5. Conclusions

In conclusion, our study confirmed that NT-proBNP both alone and combined with CITP may help identify IgAN patients at high risk for subclinical heart failure and further atherosclerotic disease. In addition, there is an effect of the modification of CKD on the association of these biomarker combinations with LVDD in HFpEF patients. A possible mechanism for the development of heart failure in CKD patients may be an elevation of central aortic systolic pressure. The pathophysiological mechanisms linking the increased prevalence of the biomarker combination with CKD, as well as its association with poor outcomes, must be investigated in larger cohorts.

### Limitations of the Study

Several limitations need to be acknowledged in our study. First, the present study was a single-center cross-sectional study with a small number of patients, which could have caused selection bias. Second, we cannot say for certain that the association between increased NT-proBNP, CITP, and decreased renal function is causal in our CKD patients. Third, the renal function was estimated, not measured, although the use of eGFR is widely accepted in the literature to describe renal function. Fourth, as a consequence, the low frequency of the biomarker combination as defined in previous studies precluded the performance of subgroup analyses in the current study. In this regard, further studies should be performed to examine the correspondence between histological and biochemical aspects of myocardial fibrosis in different HF stages. Therefore, further analyses in large and independent cohorts of patients are necessary to confirm these findings. Fifth, potential problems related to multiplicity could have influenced the findings. Finally, because they are descriptive in nature, the associations found between renal disease, circulating biomarkers, and LV dysfunction do not establish causality.

## Figures and Tables

**Figure 1 ijms-24-10336-f001:**
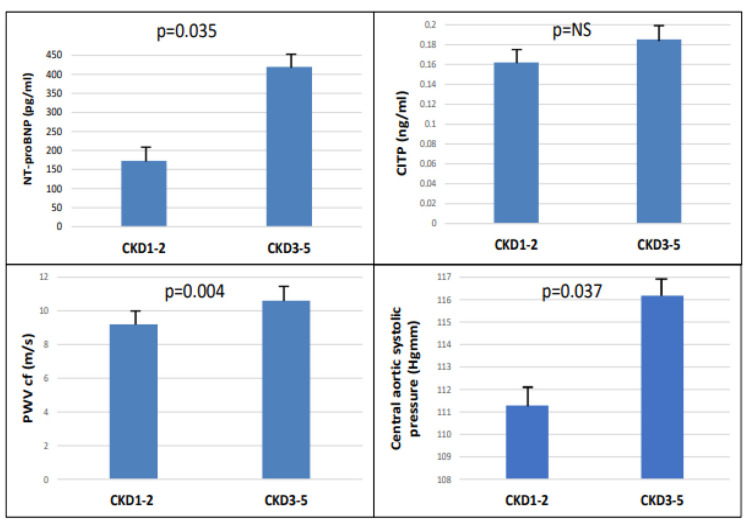
The NT-pro-BNP, CITP, PWVcf, and central aortic systolic pressure in CKD 1-2 vs. CKD 3-5.

**Figure 2 ijms-24-10336-f002:**
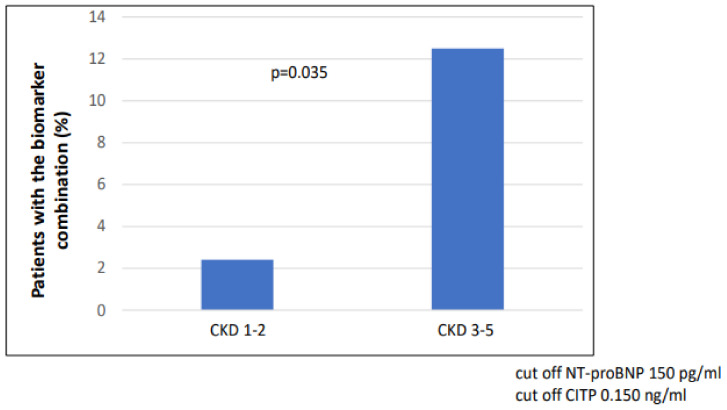
Combination of biomarker (NT-proBNP and CITP) positivity in CKD 1-2 vs. CKD 3-5.

**Figure 3 ijms-24-10336-f003:**
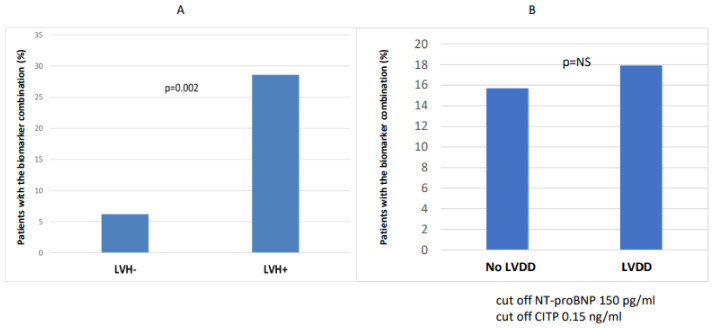
Combination of biomarker (NT-proBNP and CITP) positivity in LVH- vs. LVH+ (**A**) and with and without diastolic dysfunction (**B**).

**Figure 4 ijms-24-10336-f004:**
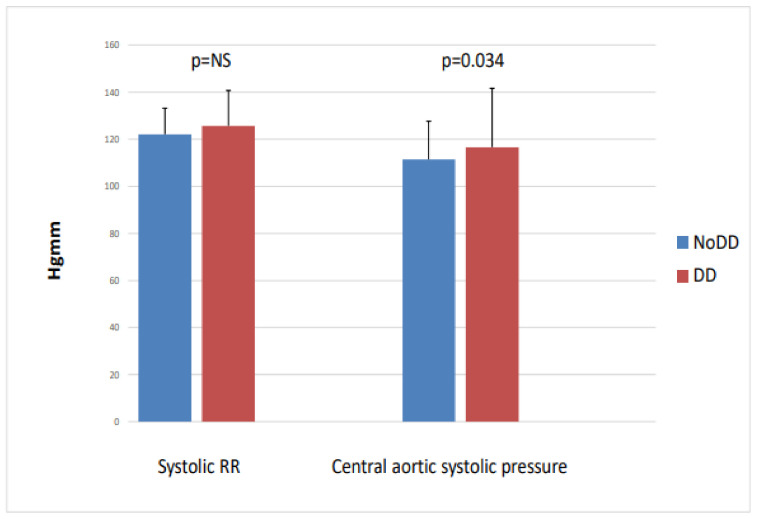
Peripheral systolic blood pressure and central aortic systolic pressure with and without diastolic dysfunction in IgAN.

**Table 1 ijms-24-10336-t001:** Baseline characteristics of IgAN patients.

Clinical Data	All Patients (n = 90)	CKD 1-2 (n = 42)	CKD 3-5 (n = 48)	*p*
Male/female (n, %)	50/40 (56/44)	25/17 (60/40)	25/23 (52/48)	NS
Age (year)	54.92 ± 14.47	49.69 ± 14.80	59.68 ± 12.69	<0.001
Systolic/diastolic blood pressure (Hgmm)	124/78 ± 13/8	122/77 ± 14/8	125/78 ± 13/7	NS
cfPWV (m/s)	9.9 ± 2.33	9.2 ± 2.18	10.6 ± 2.29	0.004
Augmentation index (%)	26.5 ± 13.68	25.7 ± 15.74	27.9 ± 11.63	NS
Aortic PP (Hgmm)	38.4 ± 6.99	35.0 ± 3.46	43.5 ± 7.94	NS
Aortic Pressure (Hgmm)	113.88 ± 12.8	111.33 ± 13,01	117.02 ± 12.72	0.044
Biomarkers				
NT-proBNP (pg/mL)	256.22 ± 404.72	173.22 ± 382.22	419.03 ± 775.07	0.035
CITP (ng/mL)	0.185 ± 0.21	0.162 ± 0.13	0.185 ± 0.25	NS
Metabolic parameters				
BMI (kg/m^2^)	28.51 ± 5.75	27.57 ± 5.21	29.34 ± 6.17	NS
Obesity (n, %)	60 (67)	25 (60)	35 (73)	NS
Hypertension (n, %)	78 (87)	28 (67)	43 (90)	0.005
Diabetes mellitus (n, %)	16 (18)	3 (7)	12 (25)	0.013
Dyslipidemia (n, %)	32 (36)	16 (38)	16 (33)	NS
Metabolic syndrome (n, %)	66 (73)	16 (38)	23 (48)	NS
Echocardiographic parameters				
LV EF(%)	63.42 ± 5.86	65.25 ± 5.39	63.18 ± 6.02	NS
LVMI (g/m^2^)	102.76 ± 27.8	100.63 ± 22.37	111.07 ± 23.21	0.022
LVM (g)	214.68 ± 62.2	234.4 ± 75.6	238.0 ± 51.26	NS
RWT	0.436 ± 0.07	0.468 ± 0.08	0.472 ± 0.07	NS
LVH (n, %)	41 (45)	16 (38)	25 (52)	NS
DD (n/%)	44 (49)	18 (43)	26 (54)	NS
E/A	1.03 ± 0.38	1.11 ± 0.39	0.98 ± 0.37	NS
Laboratory results				
eGFR (mL/min)	47.57 ± 23.24	72.61 ± 7.11	34.5 ± 16.32	<0.001
MAU (mg/L)	247.62 ± 312.78	111.32 ± 204.06	341.63 ± 345.01	<0.001
Uric acid (umol/L)	314.23 ± 83.72	300.79 ± 61.4	326.38 ± 98.92	NS
Total cholesterol (mmol/L)	4.93 ± 1.31	4.98 ± 1.36	4.89 ± 1.3	NS
HDL (mmol/L)	1.34 ± 0.42	1.42 ± 0.43	1.28 ± 0.41	NS
LDL (mmol)	2.90 ± 1.14	2.94 ± 1.21	2.85 ± 1.09	NS
TG (mmol)	1.85 ± 1.36	1.73 ± 1.47	1.97 ± 1.27	NS
Hb (g/dL)	133.71 ± 28.68	144.16 ± 14.47	124.88 ± 34.65	0.001

cfPWV: carotid–femoral pulse wave velocity; PP: pulse pressure; NT-proBNP: N-terminal prohormone of brain natriuretic peptide; CITP: carboxy-terminal telopeptide of collagen type I; eGFR: estimated glomerular filtration rate; MAU: microalbuminuria; BMI: body mass index; LV EF: left ventricular ejection fraction; LVMI: left ventricular mass index; LVM: left ventricular mass; RWT: relative wall thickness; LVH: left ventricular hypertrophy; DD: diastolic dysfunction; E/A: early and late mitral inflow diastolic velocity; HDL: high density lipoprotein; TG: triglyceride; Hb: hemoglobin; RDW: red blood cells distribution width.

**Table 2 ijms-24-10336-t002:** Correlations with NT-proBNP and CITP.

	eGFR		LVMI		Aortic PP		Central Aortic Systolic Pressure		PWVcf		Hb	
	r	*p*	r	*p*	r	*p*	r	*p*	r	*p*	r	*p*
NT-proBNP	−0.428	<0.001	0.354	<0.01	0.409	<0.001	0.325	<0.01	0.469	0.034	0.242	0.001
CITP	0.048	NS	0.295	<0.01	0.478	<0.01	0.152	NS	0.312	0.011	0.102	NS

**Table 3 ijms-24-10336-t003:** Univariate and multivariate regression analysis of NT-proBNP.

UNIVARIATE								MULTIVARIATE						
Parameter	B	Std. Error	Beta	t	*p*	Lower Bound 95% CI	Upper Bound 95% CI	B	Std. Error	Beta	t	*p*	Lower Bound 95% CI	Upper Bound 95% CI
Age	4.768	0.719	0.577	6.632	0.001	3.339	6.196	−1.819	2.789	−0.220	−0.652	0.517	−7.391	3.753
Gender	159.748	64.864	0.254	2.463	0.016	30.844	288.652	−228.633	73.573	−0.363	−3.108	0.003	−375.612	−81.654
HT	268.648	44.819	0.543	5.994	0.001	179.551	357.744	23.082	126.902	0.047	0.182	0.856	−230.434	276.598
DM	267.687	115.015	0.243	2.327	0.022	39.046	496.328	−19.403	102.033	−0.018	−0.190	0.850	−223.239	184.432
BMI	8.618	1.470	0.534	5.862	0.001	5.695	11.540	−5.238	6.455	−0.325	−0.811	0.420	−18.134	7.658
Dyslipidemia	130.828	95.219	0.148	1.374	0.173	−58.525	320.182	−166.289	82.867	−0.188	−2.007	0.049	−331.834	−0.743
eGFR	2.614	0.774	0.344	3.375	0.001	1.074	4.154	−5.673	1.527	−0.746	−3.715	0.001	−8.723	−2.622
MAU	0.453	0.123	0.385	3.688	0.001	0.208	0.697	−0.130	0.138	−0.111	−0.944	0.349	−0.406	0.145
PWVao	30.825	5.149	0.549	5.987	0.001	20.584	41.065	14.585	18.203	0.260	0.801	0.426	−21.779	50.949
DD	196.064	41.935	0.461	4.675	0.001	112.626	279.502	62.255	92.598	0.146	0.672	0.504	−122.731	247.241
LVMI	2.665	0.370	0.623	7.206	0.001	1.929	3.401	7.725	1.513	1.805	5.107	0.001	4.703	10.748

HT: hypertension; DM: diabetes mellitus; BMI: body mass index; eGFR: estimated glomerular filtration rate; MAU: microalbuminuria; cfPWV: carotid–femoral pulse wave velocity of the aorta; DD: diastolic dysfunction; LVMI: left ventricular mass index.

**Table 4 ijms-24-10336-t004:** Univariate and multivariate regression analysis of CITP.

UNIVARIATE								MULTIVARIATE						
Parameter	B	Std. Error	Beta	t	*p*	Lower Bound 95% CI	Upper Bound 95% CI	B	Std. Error	Beta	t	*p*	Lower Bound 95% CI	Upper Bound 95% CI
Age	0.003	0.001	0.638	6.980	0.001	0.002	0.004	0.001	0.002	0.242	0.556	0.580	−0.003	0.006
Gender	0.157	0.041	0.416	3.857	0.001	0.076	0.238	−0.065	0.057	−0.172	−1.143	0.258	−0.179	0.049
HT	0.182	0.028	0.616	6.537	0.001	0.127	0.238	0.037	0.098	0.125	0.379	0.706	−0.159	0.234
DM	0.155	0.076	0.235	2.020	0.047	0.002	0.307	−0.010	0.079	−0.015	−0.123	0.903	−0.168	0.148
BMI	0.006	0.001	0.625	6.704	0.001	0.004	0.008	−0.002	0.005	−0.256	−0.496	0.622	−0.012	0.008
Dyslipdemia	0.165	0.061	0.312	2.687	0.009	0.042	0.287	−0.044	0.064	−0.083	−0.685	0.496	−0.172	0.084
eGFR	0.003	0.001	0.609	6.329	0.001	0.002	0.004	0.001	0.001	0.172	0.664	0.509	−0.002	0.003
MAU	0.001	0.001	0.370	3.191	0.002	0.001	0.000	<−0.001	0.000	−0.008	−0.054	0.957	0.001	0.001
PWVao	0.021	0.003	0.615	6.379	0.001	0.014	0.027	−0.016	0.014	−0.479	−1.146	0.257	−0.044	0.012
DD	0.164	0.024	0.645	6.866	0.001	0.117	0.212	0.111	0.072	0.435	1.549	0.127	−0.033	0.254
LVMI	0.002	0.001	0.655	7.154	0.001	0.001	0.002	0.002	0.001	0.656	1.438	0.156	−0.001	0.004

HT: hypertension; DM: diabetes mellitus; BMI: body mass index; eGFR: estimated glomerular filtration rate; MAU: microalbuminuria; cfPWV: carotid–femoral pulse wave velocity of the aorta; DD: diastolic dysfunction; LVMI: left ventricular mass index.

## Data Availability

The data underlying this article cannot be shared publicly due to Hungarian regulations concerning the privacy of the individuals that participated in the study. The data can be shared on reasonable request to the corresponding author if accepted by the Regional Committee for Medical and Health Research Ethics and local Data Protection Officials.

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
