# Peer review of "The Role of Two Heart Biomarkers in IgA Nephropathy"

_ijms, 2023, doi:10.3390/ijms241210336_

Round 1

Reviewer 1 Report

Comments and Suggestions for Authors:

1. The aim of the research is missing in the Abstract and in the Introduction.

2. There are no inclusion and exclusion criteria for the study.

3. There is a mistake in the text. The authors placed the results chapter as 2, instead of the Material and Methods chapter, which they placed after the Discussion in chapter 4.

Author Response

The aim of the research is missing in the Abstract and in the Introduction.

Answer: Corrected in the abstract. 

There are no inclusion and exclusion criteria for the study.

Answer: Corrected in the Material and Methods chapter.

There is a mistake in the text. The authors placed the results chapter as 2, instead of the Material and Methods chapter, which they placed after the Discussion in chapter 4.

Answer: This is the journal's specification regarding to the order of the chapter (please see in the journal's Manuscript Preparation section)

Reviewer 2 Report

The authors performed the study very nicely. Although the correlation with eGFR and BNP is not very strong (r=0.428), the authors knows the limitations and expressed their potential interpretations based on the limitations such as sample size, and CKD estimation. 

The suggestion to authors is to clealry explain the definitions of CKD, the various indices and measurments such as Augmentation index.

Overall the study is well performed.

Author Response

The suggestion to authors is to clealry explain the definitions of CKD, the various indices and measurments such as Augmentation index.

Answer: We clearly described the definition of CKD and Augmentation index in the text.

Round 2

Reviewer 1 Report

Dear Authors,

thank you for including my comments in the manuscript, adding the purpose and criteria for inclusion and exclusion from the study.

In fact, in this journal, the Results chapter is number 2 and the Material and Methods chapter is number 4. My mistake.

regards,

Reviewer